# Comparative Study of the Antioxidant and Enzyme Inhibitory Activities of Two Types of Moroccan *Euphorbia* Entire Honey and Their Phenolic Extracts

**DOI:** 10.3390/foods10081909

**Published:** 2021-08-17

**Authors:** Oumaima Boutoub, Soukaina El-Guendouz, Ana Manhita, Cristina Barrocas Dias, Letícia M. Estevinho, Vanessa B. Paula, Jorge Carlier, Maria Clara Costa, Brígida Rodrigues, Sara Raposo, Smail Aazza, Lahsen El Ghadraoui, Maria Graça Miguel

**Affiliations:** 1Faculdade de Ciências e Tecnologia, Campus de Gambelas, Universidade do Algarve, 8005-139 Faro, Portugal; oboutoub@ualg.pt (O.B.); selguendouz@ualg.pt (S.E.-G.); mcorada@ualg.pt (M.C.C.); sraposo@ualg.pt (S.R.); 2Laboratory of Functional Ecology and Environment, Faculty of Science and Technology, BP 2202, University Sidi Mohamed Ben Abdallah, Fez 30000, Morocco; lahsen.elghadraoui@usmba.ac.ma; 3Laboratório HERCULES, Instituto de Investigação e Formação Avançada, Universidade de Évora, Largo Marquês de Marialva 8, 7000-809 Évora, Portugal; anaccm@uevora.pt (A.M.); cmbd@uevora.pt (C.B.D.); 4Departamento de Química, Escola de Ciências e Tecnologia, Universidade de Évora, Rua Romão Ramalho 59, 7000-671 Évora, Portugal; 5Mountain Research Center (CIMO), Polytechnic Institute of Bragança, Campus Santa Apolónia, 5300-253 Bragança, Portugal; leticia@ipb.pt (L.M.E.); vanessapaula@ipb.pt (V.B.P.); 6Centre of Marine Sciences (CCMAR), Gambelas Campus, University of the Algarve, 8005-139 Faro, Portugal; jcarlier@ualg.pt; 7CIMA—Centre for Marine and Environmental Research, FCT, Campus de Gambelas, Universidade do Algarve, 8005-139 Faro, Portugal; bgrodrigues@ualg.pt; 8Laboratory of Phytochemistry, National Agency of Medicinal and Aromatic Plants (ANPMA), BP 159, Principal, Taounate 34000, Morocco; aazzasmail@ymail.com; 9Mediterranean Institute for Agriculture, Environment and Development, Campus de Gambelas, Universidade do Algarve, 8005-139 Faro, Portugal

**Keywords:** honey, phenolic compounds, antioxidant activity, *Euphorbia resinifera*, *Euphorbia officinarum*, phenolic profile

## Abstract

Honey is a natural food product very famous for its health benefits for being an important source of antioxidant and phenolic compounds. *Euphorbia* honeys obtained from different regions of Morocco were evaluated for their ability to inhibit acetylcholinesterase, lipoxygenase, tyrosinase and xanthine oxidase activities. Their antioxidant properties were evaluated using the: 2,2-diphenyl-1-picrylhydrazyl (DPPH) radical-scavenging capacity, nitric oxide scavenging activity (NO) and scavenging ability of superoxide anion radical. Then, the phenolic extracts of the same entire honey samples were evaluated by liquid chromatography coupled to diode array detection and mass spectrometry (LC-DAD-MS) and tested for the biological activities previously evaluated on the entire honeys, in order to conduct a comparative study between both (honey and phenolic extracts). The chromatographic profiles for the studied *Euphorbia* honey extracts were different. Phenolic compounds gallic acid, 4-hydroxybenzoic acid and *p*-coumaric acid were detected in all samples, whereas kampferol was only present in two samples. Physicochemical parameters and total phenolic content were also determined. Entire honey that recorded the highest rate of phenols was sample M6 (*E. resinifera*) = 69.25 mg GAE/100 g. On the other hand, the phenolic extracts had better antioxidant and enzyme inhibitory activities than the entire honeys, regardless the monofloral honey type. In conclusion, the studied *Euphorbia* honeys may have a great potential as antioxidant, anti-inflammatory and anti-tyrosinase sources for pharmaceutical and cosmetic applications.

## 1. Introduction

The Green Morocco Plan has permitted a sustainable development of the agricultural sector of this country contributing to increase its added value. The country has established programs in order to leave the subsistence farming, focusing on the promotion of the specific products of the territories, identifying hundred of local products and, consequently, labeling them under Geographical Indications, Designations of Origin or Agricultural Labels. These labeled products include fresh and dried fruits, medicinal and aromatic plants, olive and argan oils and animal origin products such as honey [1]. Three labeled honeys were registered, being two of them of *Euphorbia* monofloral origin: geographical indication “honey of desert *Euphorbia*” and protected geographical indication “*Euphorbia* honey of Tadla-Azilal” [2]. The organoleptic characteristics for the former are the color dark amber, dry herbal, wax and spicy taste, and a permanent, intense and prickly aftertaste; and for the last one the color is dark golden, bitter and peppery at the throat taste [2]. In both cases, the plant species is not provided although three monofloral *Euphorbia* honeys can be found in Morocco: *E. officinarum* subsp. *echinus*, *E. regis-jubae*, and *E. resinifera*. *E. resinifera* is an endemic species of Morocco mainly distributed in Azilal and Beni Mellal regions (Middle Atlas), whereas *Euphorbia officinarum subsp. echinus* and *Euphorbia regis-jubae* can be found in the south-western region [3].

Beyond the nutritive aspect of honeys, *Euphorbia* honey has been target of study due to its economical importance in Morocco, thereby investigation has increased on the melissopalynological and physico-chemical characterization [4,5,6,7,8] and biological properties with potential application on human health [9,10,11]. Anti-inflammatory, analgesic and antimicrobial properties have been attributed to *Euphorbia* honeys along with the ability to promote wound healing, nevertheless, the majority of works has been focused on the anti-microbial activity [9,10,11], and much less, antioxidant activity [12]. More recently, our team compared the antioxidant activity and some enzyme inhibitory activities of *Euphorbia officinarum*, *E. resinifera* plants with those of the respective two monofloral honeys [13]. This work permitted detected relative high amounts of Al, Cu and Fe in *E. officinarum* honey, suggesting environmental pollution and/or inadequate storage of honey.

In continuation of our studies, the present work aims to evaluate the antioxidant, anti-inflammatory, anti-acetylcholinesterase, anti-tyrosinase and anti-xanthine oxidase activities of seven entire *E. resinifera* and *E. officinarum* honeys from Morocco as well as their phenolic extracts for better understanding if the activities are due to the entire honey and/or to their secondary metabolites.

## 2. Materials and Methods

### 2.1. Honey Samples

Seven (n = 7) unifloral honey samples of *Euphorbia* sp. were acquired directly from the beekeepers between June and July 2018. These samples were stored in dark at room temperature until time of testing, no more than two months after collection. The Figure 1 represents the regions of harvest and the palynological classification of the samples.

### 2.2. Melissopalynological Analysis

Honey samples pollen qualitative and quantitative analysis was performed according to the International Commission for Bee Botany (ICBB) [14].

### 2.3. Physico-Chemical Parameters of Honey

Free acidity, lactonic acidity, total acidity, pH, ash content, electrical conductivity, moisture, proline content, diastase activity, HMF content, reducing sugars have been determined according to the methods used by Bogdanov [15].

#### 2.3.1. Free Acidity

The sample solutions were neutralized with a standard solution of sodium hydroxide (0.1 M) to pH 8.30 using a potentiometer combined with glass electrode (Thermo Electron Corporation, Orion 3 STAR; Beverly, MA, USA).

#### 2.3.2. Lactonic Acidity

The lactonic acidity was determined by adding the excess of sodium hydroxide (0.05 M) to the honey solution and back titrated with sulfuric acid (0.025 M), using a potentiometer combined with glass electrode (Thermo Electron Corporation, Orion 3 STAR; Beverly, MA, USA).

#### 2.3.3. Total Acidity

Total acidity is obtained by making the sum of free acidity plus lactonic acidity.

#### 2.3.4. pH

The pH was measured using a pH-meter with glass electrode (Thermo Electron Corporation, Orion 3 STAR; Beverly, MA, USA) from the honey solution prepared in CO_2_-free distilled water.

#### 2.3.5. Ash Content

Five grams of the honey sample were put in a porcelain dish and burned in a temperature between 350–400 °C in electric furnace for at least 2 h, after cooling the porcelain dish was placed in desiccator and weighted.

#### 2.3.6. Electric Conductivity

This parameter was obtained by using a conductivity meter (Thermo Electron Corporation, Orion 3 STAR; USA) in the aqueous honey solution.

#### 2.3.7. Water Content (Moisture)

Water content (moisture %) was determined with a Abbe refractometer (HANNA HI96801, HANNA Instruments, Nușfalău, Romania) at 20 °C and using the Wedmore’s Table.

#### 2.3.8. Proline Content

For proline content, 5 g of honey was dissolved in 100 mL of distilled water in a volumetric flask. To carry out this test it is necessary to have three test tubes for each sample. The first tube (blank test) contains 500 µL of distilled water mixed with 1 mL of formic acid (H.COOH) (98%) and 1 mL ninhydrin solution (3%) in ethylene glycol monomethyl ether. The second tube (sample test), 500 µL of honey solution was mixed with 1 mL of formic acid (98%) and 1 mL of ninhydrin solution (3%). The last test tube contains 500 µL of proline solution standard (0.8 mg/25 mL) mixed with 1 mL of formic acid (98%) and 1 mL of ninhydrin solution (3%). The three tubes were shaked for 15 min and put in a water bath for 15 min at 70 °C, by the end 5 mL of 2-propanol 50% were added and leave it to cool for 45 min, the absorbance was measured at λ = 510 nm, using a spectrophotometer Shimadzu 160-UV (Shimadzu Europe GmbH, Duisburg, Germany).

#### 2.3.9. Diastase Activity

Diastase activity (Shade units/g) was determined by weighing 10 g of honey dissolved in 15 mL and 5 mL of acetate buffer (pH 5.3), the mixture was transferred to 50 mL volumetric flask containing 3 mL of sodium chloride solution (2.9 g/100) and adjust the volume to the mark with water. Ten mL of this solution was measured and introduced into a 50 mL of flask and left at 40 ºC in a water bath along with a second flask with the same volume of starch solution. After 15 min, 5 mL of starch solution is pipetted into the honey solution. At periodic intervals, 0.5 mL of the mixture is pipetted for other tube and 5 mL of diluted iodine solution, and a volume water previously determined and the absorbance read at λ = 660 nm.

#### 2.3.10. Hydroxymethylfurfural (HMF)

Hydroxymethylfurfural was determined by clarifying honey samples with Carrez solution (I and II) with a sodium bisulphite solution (0.20 g/100 g) and the absorbance was measured at λ = 284 nm.

#### 2.3.11. Reducing Sugars Percentage

Reducing sugars percentage in each honey sample were determined with the titration in 2-time assay (preliminary assay and definitive assay) using the solutions: 5 mL of A and 5 mL of B. Fehling solution A: 69.28 g of copper sulphate pentahydrate (CuSO_4_•5H_2_O) in 1000 mL distilled water; Fehling solution B: 346 g of sodium potassium tartrate (C_4_H_4_NaO_5_•4H_2_O) and 100 g of sodium hydroxide (NaOH) in 1000 mL distilled water. Five mL of each solution were measured into an Erlnmeyer flask and from a burette, the honey solution was left drop until the upper phase remains colourless.

### 2.4. Estimation of Honey Colour

The colour of honey was evaluated as reported by Aazza et al. [16]. A solution of 1 g in 2 mL of distilled water was prepared and the absorbance measured at λ = 635 nm (A**_635_**). The colour was calculated using the equation:mm Pfund = −38.7 + 371.39 × A**_635_**

### 2.5. Determination of Mineral Elements of Honey Samples

The mineral elements quantified were Fe, Zn, Mn, Cu, Al, Ca, K, Mg, Na. Five grams of honey were calcined at 550 °C and after cooling 5 mL of nitric acid 0.1 M were added, shaked and heated until completely dry on a hot plate. Ten mL of nitric acid 0.1 M was added and make up to 25 mL with distilled water [17]. For the Ca, Mg, Mn, Zn, Cu, and Fe, the measurement was done through flame atomic absorption spectroscopy air-acetylene using a novAA 350 (Analytik Jena, Jena, Germany), while for the analysis of Na, K and Al, microwave plasma atomic emission spectroscopy (4200 MP-AES, Agilent, Santa Clara, CA, USA) was used. The concentrations were expressed as mg/ kg honey.

### 2.6. Carbohydrate Content of Honey Samples by High Performance Liquid Chromatography (HPLC)

The sugar quantification was done using a chromatograph Hitachi LaChrom Elite HPLC, Japan, equipped with a refractive index detector (Hitachi L-2490, Tokyo, Japan). The col-umn used was a Purospher STAR NH_2_ (5 μm particle size) (Merck, Darmstadt, Germany). The separation of carbohydrates was achieved with an isocratic elution having as mobile phase acetonitrile and water (85:15, *v/v*), at room temperature. The preparation of honey samples and the quantification of carbohydrates was done according to the methodology previously reported [3].

### 2.7. Samples Extraction for the Determination of Phenolic Compounds

This study relates to the comparison between the seven *Euphorbia* entire honeys and their phenolic extracts. Accordingly, 5 g of each entire honey sample was diluted in 10 mL of distillated water in the day of the assay in order to study the antioxidant and enzymatic inhibitory activities. On the other hand, the extraction of phenolic compounds for the seven honey samples was performed according to procedures previously described [18]) with slight modifications: 10 g of each honey sample was dissolved in 50 mL of distilled water and mixed well for 5 min using a vortex. In each flask solution, 50 mL of ethyl acetate were added and were placed for 45 min in a shaker (1700 rpm) (Edmund Bühler, TH 15, Bodelshausen, Germany). The mixture was then transferred to a separating funnel for separation of the phases. The liquid-liquid extraction was repeated three more times. Then, the combined ethyl acetate extracts were evaporated using a vacuum rotary evaporator (Heidolph, 94200, BIOBLOCK SCIENTIFIC, Schwabach, Germany) at 36 °C, after that the residue was collected in methanol (5 mL). The phenol content and their identification as well as the antioxidant and enzymatic activities were subsequently determined for each phenolic extract.

#### 2.7.1. Total Phenol Content

The total polyphenol content in honey as well as in the methanolic extract were determined as stated by Boutoub et al. [13]. The total polyphenol content was expressed as mg gallic acid equivalents per 100 g (mg GAE/100 g).

#### 2.7.2. Liquid Chromatography Coupled to Diode Array Detection and Mass Spectrometry (LC/DAD/MS) for the Identification of Phenolic Compounds

Five µL of the methanolic extracts M1–M7 prepared in step 2.7. (‘Samples extraction for the determination of phenolic compounds’) were analysed by LC/DAD/MS technique. The extracts were previously filtered using a 0.45 µm PTFE syringe filter.

A mass spectrometer (LCQ Fleet, Thermo Finnigan, San Francisco, CA, USA), equipped with an electrospray ionization (ESI) source and an ion trap mass analyser was used. MS conditions for the analysis were the following: 300 °C for capillary temperature, 5.0 kV for source voltage, 100.0 μA for source current and −3.0 V for capillary voltage, in negative ion mode.

Samples were analysed both in full MS mode (*m/z* 50–800), and in Selective Reaction Monitoring (SRM) mode (normalized collision energy of 30%). Selected reactions are presented in the Section 3.6 and were optimized using standards available in the laboratory.

The MS equipment was coupled to an LC system (Surveyor, Thermo Finnigan, San Francisco, CA, USA) with an autosampler and a diode array detector (DAD). A reversed-phase analytical column was used (Zorbax Eclipse XDB C18, Rapid Resolution, particle size 3.5 μm, 150 mm × 4.6 mm, Agilent Technologies, Santa Clara, CA, USA). Tray temperature was set at 24 °C and column temperature was set at 30 °C. Chromatographic separation was performed at a flow rate of 0.2 mL/min. The mobile phase was composed by 0.1% (*v/v*) formic acid aqueous solution (A) and acetonitrile (B) with the following elution programme: 0.0% B (0–5 min), 0.0–10.0% B (5–20 min) and 10.0–50.0% B (20–60 min). DAD detector was programmed to acquire data from 200 to 800 nm.

### 2.8. Antioxidant Activity

#### 2.8.1. 2,2-Diphenyl-1-picrylhydrazyl (DPPH) Scavenging Activity

Free radical scavenging activity determination by DPPH scavenging activity was performed as described by Boutoub et al. [13]. The volumes used for phenolic extract and honey samples were 25 and 200 μL, respectively. The percentage of inhibition was determined using the formula: Percentage of inhibition = [(A_0_ − A_1_)/A_0_ × 100]; with A_0_ representing the absorbance of the control and A_1_ the absorbance of the sample. The sample concentration providing 50% inhibition (IC_50_) was achieved by plotting the inhibition percentage against extracts concentrations.

#### 2.8.2. Nitric Oxide Scavenging Activity

The nitric oxide (NO) scavenging activity was carried out according to Boutoub et al. [13]. In both phenolic extract and honey sample the volume used was 150 μL. The IC_50_ values were obtained as aforementioned.

#### 2.8.3. Scavenging Ability of Superoxide Anion Radical

Scavenging ability of superoxide anion radical was assayed as reported by Boutoub et al. [13]. In both phenolic extract and honey sample the volume used was 25 μL. The IC_50_ values were obtained as aforementioned.

### 2.9. Enzymatic Activities

#### 2.9.1. Inhibition of Acetylcholinesterase

The acetylcholinesterase inhibition was carried out with few modifications as reported by Boutoub et al. [13]. For phenolic extract 25 µL were used while for honey sample the volume was 300 µL. The percentage of inhibition of acetylcholinesterase activity was determined and the (IC_50_) value was calculated.

#### 2.9.2. Inhibition of Lipoxygenase

The lipoxygenase assay is used as an indicator of anti-inflammatory and antioxidant activity [19]. The inhibition action of honey solution and plant extract was determined as reported by Boutoub et al. [13] with some modifications. In short, 25 µL of phenolic extract were used while for honey solution, the volume was 150 µL. The results were expressed as IC_50_ value.

#### 2.9.3. Inhibition of Tyrosinase

The tyrosinase activity was determined based on the protocol reported by El-Guendouz et al. [20] with slight modification. The total assay mixture consisting on 50 µL of honey solution and 25 µL of phenolic extract was used for this activity. The results were expressed as IC_50_ value.

#### 2.9.4. Inhibition of Xanthine Oxidase

The inhibitory activity of seven phenolic extracts and their honey solution was determined as described by El-Guendouz et al. [19], but using 25 µL of methanolic extract and 150 µL of honey solution. The results were expressed as IC_50_ value.

## 3. Results and Discussion

### 3.1. Pollen Analysis

The microscopic analysis of the sediment (Table 1) showed that the predominant pollen grains of the two samples (M2 and M6) from Beni Mellal-Khénifra region was *Euphorbia resinifera*, with a percentage of 48% for M2 and 45% for M6, being both classified as monofloral. These findings are in accordance to those recently reported [7], in which lower than 50% of *E. resinifera* pollen could be found in more than 20 samples studied. The second most prevalent pollen types were *Caesalpinia pulcherrima* (21.80%) and *Genista hirsuta* (12.83%) in M2 and M6 honeys, respectively. Moreover, *Caesalpinia pulcherrima* pollen was absent in sample M6. In contrast to M2, in M6 sample, pollen of *E. officinarum* was found (5.93%). These accompanying pollen species are not the same previously reported [7].

Regarding the other five samples (M1, M3, M4, M5, M7), the analysis showed the dominance of the pollen species of *E. officinarum*. The M7 sample from the Souss-Massa region (Table 2), more exactly Tiznit (Figure 1), recorded a value of 42.22% of the predominantly *E. officinarum* pollen species, on the other hand the M1 sample collected in the region of Guelmim-Oued Noun and closer to Ait Baamrane (Figure 1), recorded a value of 55.70% as the highest percentage of *E. officinarum* pollen. The second pollen types comprised between 10.3 and 17.74% were from *Quercus rotindifolia, Caesalpinia pulcherrima, Caesalpinia spinosa, Quercus suber*, and *Pinus pinaster* (Table 2) sequentially regarding samples M1, M3, M4, M5 and M7.

### 3.2. Physicochemical Parameters

The results obtained for the physicochemical parameters are summarized in Table 2. The pH values ranged from 3.75 to 4.08 with an average value of 3.86 to the seven samples. Our results are in agreement with those of Moujanni et al. [4], generally the pH value of *E. resinifera* honey is slightly higher than that of *E. officinarum* honey.

The acidity of honey is due to the presence of organic acids, produced from nectar during the maturation by glucose oxidase [4], and also to the organic acids such as gluconic acid and their lactones and esters [6]. Concerning the free acidity values (Table 3), they ranged from 10.08 to 19.04 meq/kg, all values below the Codex Alimentarius Commission [21] limit of tolerance (50 meq/kg). The results obtained were lower comparing with the results found by Bettar et al. [6] with values ranging from 16 to 80 meq/kg.

The moisture content in the honey samples was between 18.62% and 20.00% (Table 2), being within the limit (≤ 20%) recommended by the international quality regulations [21]. The samples of *E. resinifera* had lower percentages of moisture (M2 = 18.62% and M6 = 18.73%) than the five remaining samples of *E. officinarum*. The values found in *E. resinifera* honey are within the mean values referred by Chakir et al. [8] (17.06%). Regarding the results of *E. officinarum*, there was also an agreement with the values found by Bettar et al. [6] (19.60–21.70%). These differences in moisture percentages can be attributed to climatic conditions [22] where honey samples were collected.

The diastase activity is an indicator of the freshness and the detection of heat induced defects and the improper storage of honey [22]. Diastase activity shows values between 11.43 (M4) and 115.89 (M6) Shade units/g. The results were within the values found for the totality of *Euphorbia* honeys samples tested by Chakir et al. [8].

The hydroxymethylfurfural (HMF) is an important criterion to evaluate storage time and the heat damage [6]. In general, fresh honey does not contain or contain very low or trace amounts of HMF. In this study, all *Euphorbia* honey samples showed HMF values ranging between 2.29 to 80.48 mg/kg, being almost all samples within the limit established by the Codex Alimentarius Commission (80 mg / kg) [21].

Electrical conductivity of honey is related to the concentration of mineral and organic acids and dependent on the floral origin [8]. In our results, the values found for the electrical conductivity were between 342.30 and 553.67 µS/cm, where *E. officinarum* samples recorded the higher values (M1 = 553.67 µS/cm). Our results are similar to those reported by other authors (561.18 µS/cm in *E. officinarum*) [8]. Concerning the *E. resinifera* honey the values were:M2 = 379.33 µS/cm and M6 = 455.33 µS/cm. Previous results [8] obtained a value of 410.62 µS/cm for *E. resinifera*, quite similar to our results. In this study, honey colour was between 103.67 mm Pfund (M7) (dark) and 510.96 mm Pfund (M6), corresponding to dark-amber colour.

In our study, the ash content values varied between 0.13 % and 0.19%. According to the European legislation [23] the value of ash content must not exceed 0.6% which means that our samples respect the proposed standards. In this study we did not find a positive correlation between the values of ash percentage and electrical conductivity, as it had been elsewhere reported [17].

Honey contains several amino acids, being proline the major one [24]. This parameter indicates honey maturity. In our study, the higher values were for *E. resinifera* honey M6 = 1485.71 mg/kg and the lowest one was for *E. officinarum* honey M1 = 584.96 mg/kg (Table 2), and both of them contained more than the minimum acceptable proline concentration 200 mg/kg for honey samples [25].

Glucose and fructose are the main sugars present in honey samples. The reducing sugars, were significantly higher in *E. resinifera* samples (66.67% and 70.67%) than in the monofloral *E. officinarum* honeys (Table 2). In both cases, they are higher than the minimum required levels of 60% by the Codex Alimentarius Commission. Our results suggest that the reducing sugars’ percentage may be related to the floral origin of honeys.

With very few exceptions, there were not significant differences in the physicochemical parameters between the two monofloral honeys. Similar results were already reported [6] but between *E. officinarum* and *E. regia-jubae*.

### 3.3. Sugar Profile and Quantification

Fructose and glucose are the main carbohydrates of honey [26]. In all samples fructose (31.53 g/100 g–39.48 g/100 g) was at higher concentration than glucose (27.77 g/100 g–34.17 g/100 g) (Table 3). It was not possible to detect differences between *E. resinifera* and *E. officinarum* honeys. The values for sucrose ranging from 3.45 and 5.37 g/100 g (Table 4). Other sugars in lower quantities was found as turanose, maltose and trehalose. These results are concordant with previous works [4,13], for the different sugars presented for *E. resinifera* and *E. officinarum* honeys.

### 3.4. Elemental Mineral Analysis

Contents of each mineral element found in the seven honeys expressed in mg/kg (fresh weight) are shown in Table 4. The potassium (K) was the most important mineral element with an average content of 409 mg/kg. *E. officinarum* honeys had the highest K content (M5 = 533 mg/kg) but also the lowest one (M7 = 324 mg/kg). Concerning the *E. resinifera* honeys, M6 had the highest value of potassium 503 mg/kg and M2 = 345 mg/kg shows the lowest content. Concerning the second most important mineral element for both species, calcium (Ca), *E. officinarum* samples, M1 records the highest value (153 mg/kg) and M3 shows the smallest quantity (70.93 mg/kg). For *E. resinifera* M2 presented the highest calcium value (115 mg/Kg), followed by M6 with 103 mg/kg. These results agree with those already reported by Elamine et al. [17], the two highest minerals found were the potassium in the first range followed by the calcium, which agrees with our results, nevertheless there was not possible to detect differences between the two monofloral honeys (Table 4).

The third most important mineral element was sodium (Na), nevertheless the values differed significantly, ranging from 34.96 to 125.05 mg/kg (Table 4). Such variability was also found by Bettar et al. [6] for *E. officinarum* honey and by Moujanni et al. [4] for *E. resinifera* honey.

### 3.5. Total Phenolic Content

Total phenolic content of the entire honey samples and their phenolic extracts are depicted in Table 5. Polyphenols are present in small amount and derived from the pollen of the plant foraging by the honeybee [27]. The highest total phenolic levels were detected in the entire honeys and the lowest in their phenolic extracts (Table 5). These results are in opposite to that previously reported [28], but similar to that found by Ferreira et al. [29] for Portuguese honeys. In addition, these authors also reported a correlation between the total phenol content in the entire honeys and their colour. In our case, the darker entire honey sample (M6) had the higher total phenolic content contrariwise the amber honey sample (M7) that had the lower total phenolic content.

Another study [30], reported that the entire honey contains some non-phenolic reducing compounds contributing to increase the absorbance values (interferences) in the total phenol assay, which gives erroneous values on the rate of phenol present in the sample. In fact, honey contains reductive sugar or organic acids, and these compounds interfere and may cause the values found of total phenols, determined by Folin-Ciocalteu’s method, in entire honey samples [30].

### 3.6. Phenolic Profile of Honey Extracts

M1-M7 honey extracts chromatographic profiles are depicted in Figure 2 and Table 6. Nineteen phenolic compounds’ standards were tested for SRM analysis, but some of them were not detected in honey extracts, and therefore were not included in the results’ table. In a general way, the chromatographic profiles for the studied *Euphorbia* honey extracts were quite different. Phenolic compounds gallic acid, 4-hydroxybenzoic acid and *p*-coumaric acid were detected in all samples, although in different ratios. Naringenin was identified in all but sample M5.

Abscisic acid was detected in 5 of the 7 studied honey samples, being the major compound detected in honey M6. As a plant hormone, abscisic acid is responsible for regulating plant development, growth, and response to stress. It is a common compound in honey and has been proposed as a marker for checking adulterated honey and for quality control, in *Acacia* and *Erica* honey samples [31,32], although this type of pollen had not been detected (Table 1).

### 3.7. Antioxidant Activity

#### 3.7.1. Scavenging DPPH Free Radicals

The antioxidant properties of honey samples were determined using both entire honey and their phenolic extracts (Table 7). All samples had the capacity to reduce the stable violet DPPH radical to yellow DPPH-H, with the 50% of the reduction values (IC_50_) ranging from M5: IC_50_ = 16.30 mg/mL, as the best values of *E. officinarum* entire honey, to M2: IC_50_ = 80.13 mg/mL) for *E. resinifera* entire honey [13]. Concerning the phenolic extract, the IC_50_ values ranged from PE5: IC_50_ = 2.58 mg/mL) to PE3: IC_50_ = 17.64 mg/mL). According to these results, the best antioxidant capacity (the smaller value of IC_50_ = 2.58 mg/mL) was recorded in the phenolic extracts and not in entire honey samples. This agrees with previous authors [28], who have found that the best antioxidant activities were attributed to the phenolic extract. Honey contains many biologically active compounds able to counteract the action of antioxidants, such as polyphenols [27]. The phenolic extract (PE5) presented the higher content of phenols (30.77 mg GAE/100 g) and the best antioxidant activity (IC_50_ PE5 = 2.58 mg/mL); this correspondence had already been drawn up by Rostislav et al. [26].

#### 3.7.2. Nitric Oxide (NO) Scavenging Activity

All the assessed samples were able to scavenge NO free radicals and as for DPPH free radicals, phenolic extracts were also better NO scavengers than the respective entire honeys (Table 7). Moreover, the presented results are comparable with a previous publication [12]. Indeed, in this study, the authors have found that NO scavenging activity of *E. officinarum* expressed as IC_50_ was 95.14 mg/mL. In contrast to the DPPH free radicals, NO has ubiquitous presence in the living body and is important in the maintenance of health, nevertheless at high concentrations becomes harmful [33]. The capacity for scavenging this reactive nitrogen species may reveals interesting and in the present work, all honey samples showed this activity, mainly due to the compounds of the honey extracts.

#### 3.7.3. Superoxide Anion Radical Scavenging Ability

All the tested samples presented a high capacity for scavenging superoxide radical anion (Table 7). With the exception of M3 honey sample, that had better activity than the respective extract, in the remaining samples, the extracts presented better superoxide radical anion scavenging activity than the whole honey samples, as observed for the other antioxidant activities aforementioned. This characteristic observed in this antioxidant test suggests that the responsible compounds were the phenolic compounds and not the other compounds present in the whole honey. Other authors [34], also demonstrated that phenolic compounds possess potent effects on antioxidant effects due to the ability for scavenging superoxide anion radicals. At the same time, [35] affirms that the polyphenolic contents in honey are linked to high superoxide scavenging activity. The Zantaz honey (*Bupleurum spinosum* pollen as predominant species) from Morocco [35], had recorded an IC_50_ values of 50.91 mg/mL for the superoxide activity, a very low activity compared to that found in our *E. officinarum* honey M7:IC_50_ = 1.95 mg/mL.

### 3.8. Enzymatic Activities

#### 3.8.1. Inhibition of Acetylcholinesterase Activity

The entire honeys and their extracts studied in this work were characterized by the ability to inhibit acetylcholinesterase enzyme (Table 8). Acetylcholinesterase (AChE) is a specific enzyme which breaks down the acetylcholine, a neurotransmitter in the nerve synapse [36]. A great inhibitory activity was observed in the entire honey samples (IC_50_ values from M7: IC_50_ = 3.90 mg/mL to M3: IC_50_ = 164.49 mg/mL, but not as much as their phenolic extract. *E. officinarum* is the honey which recorded the best enzymatic inhibition activity not only for its pure honey but also for the phenolic extracts as compared with the *E. resinifera* honey samples. The activities found can be partially attributed to the total phenolics. Besides, our results show that M7: IC_50_ = 3.90 mg/mL honey sample, which recorded the best anti-acetylcholinesterase activity, possessed a value of M7 = 46.14 mg GAE/100 g as the rate of total phenol as well as better scavenging ability of superoxide anion radical (IC_50_ = 1.95 mg/mL). Concerning the phenol extract, the same trend was observed. Similar behaviour was noticed for the PE5: IC_50_ =1.13 mg/mL which possesses the best AChE inhibition activity, the higher TPC (PE5 = 30.74 mg GAE/100 g) and the best DPPH antioxidant activity (PE5: IC_50_= 2.58 mg/mL). As indicated by some authors [37,38], the ability of honey and their methanolic extracts to inhibit the AChE enzyme is related to the presence of a higher total phenol content and its antioxidant potency.

#### 3.8.2. Inhibition of Lipoxygenase Activity

This study determined the anti-inflammatory activities of various samples of Moroccan *Euphorbia* honeys and their methanolic extracts through the anti-lipoxygenase activity. The results of the entire honeys samples and their phenolic extracts are summarized in Table 8. The anti-lipoxygenase activity was observed to vary depending on the type of *Euphorbia* species source. The *E. resinifera* entire honeys (M2 and M6) had the highest capacity for inhibiting the lipoxygenase activity. Concerning the respective phenolic fractions, only PE2 was also better than the remaining phenolic fractions, although PE6 also presented high ability but along with PE7 (*E. officinarum*). The inhibition of lipoxygenase has been considered as an indicator of anti-inflammatory and antioxidant activities [20] because lipoxygenase-generated free radicals disrupt membrane selective permeability, through peroxidation of membrane phospholipids [39]. Samples having high total phenol content and antioxidant activity will surely have an anti-inflammatory activity [39].

#### 3.8.3. Inhibition of Tyrosinase Activity

The tyrosinase inhibitory activity of honey and the respective honey methanolic extracts is shown in Table 8. The highest inhibition activity of honey was found in *E. resinifera* and respective extract (PE2). Moreover, this sample is the only that entire honey has higher activity than the respective phenolic fraction (Table 8). Tyrosinase is well-known as a key enzyme in melanin biosynthesis [40]. Melanin production in human skin represents a primary defense mechanism against the UV light, but the excessive accumulation and formation of epidermal pigmentation can cause various disorders [40], therefore the capacity of inhibition of tyrosinase enzyme that a natural product possess can be useful. According to Akin et al. [41], the antioxidant activity of natural sources plays a very important role in the inhibition of tyrosinase. For example, PE2 had good ability for scavenging superoxide radical anions (IC_50_ = 0.95 mg/mL) and also for inhibiting tyrosinase activity (IC_50_ = 15.28 mg/mL) According to Petrillo et al. [40] Sardinian honeys showed the highest anti-tyrosinase activity (IC_50_: 64.3 ± 1.6 mg/mL), lower as compared to *Euphorbia* honey sample (M2 and PE2) of the present work, nevertheless closer to the remaining samples.

#### 3.8.4. Inhibition of Xanthine Oxidase Activity

The entire honey showed a percentage of inhibition ranging from M7 = 48.78% to M3 = 94.92% and the phenolic extracts from PE7 = 48.58 to PE3 = 99.66%. *E. officinarum* pure honey and phenolic extract exhibited the highest percentage of inhibition: >99% for both of them. According to some authors [42], honey is an important inhibitor of xanthine oxidase depending on the floral source and the phenolic contents. Moreover, the ROS are generated by the reaction catalyzed by xanthine oxidase, which catalyzes the oxidation of hypoxanthine to xanthine and at the end the uric acid, responsible for several diseases [40]. The blocking of uric acid production is based on blocking the key enzyme, xanthine oxidase [43], and according to several studies this blocking can be induced by substances with a high antioxidant power such as the phenolic compounds. In this work, *E. offinarum* honey and phenolic extracts showed a varied antioxidant power (Table 7) and phenolic content (Table 4), which gives them the property to block the formation of uric acid from derivatives obtained by xanthine oxidase enzyme.

### 3.9. Correlations between Antioxidant Activities and Enzymatic Activities

As shown in Table 9, the phenolic compounds were strongly negatively correlated with the nitric oxide scavenging activity of entire honey samples (r = −0.56, *p* <0.01), that is, higher phenolic content, lower the IC_50_ values, therefore better activity. A correlation between phenolic content and the remaining activities were not observed. On the other hand, the IC_50_ values of DPPH and NO free radical scavenging activities negatively correlated with total phenolics of the extracts but the anti-lipoxygenase and anti-tyrosinase positively correlated with phenolic content of the extracts. This suggests that the antioxidant activity of the phenolic fraction enhances with the increase of phenolic content but playing an inverse role on the enzymatic inhibitory activities.

This study shows that the antioxidant capacity of *Euphorbia* honey comes from the phenolic compounds present in whole honey, with the exception of superoxide anion radicals scavenging activities were no correlation could be found. On the other hand, the phenolic content has a negative effect on the enzymatic inhibitory activities since a positive correlation was observed between the phenolic content and the activities found.

## 4. Comparison of Antioxidant Activities and Enzymes’ Inhibitory Activities with Other Moroccan Honeys

The antioxidant activities and other biological properties of Moroccan honeys from diverse botanical source have been studied. Concerning the antioxidant activity, several methods have been used predominating the ability for scavenging some type of free radicals, such as DPPH, superoxide and nitric oxide (Table 10). In this Table are those results obtained from the same type of assay used in the present work, in order to better compare the results. Moreover, in this Table only are depicted the lowest and the highest IC_50_ values found by the authors although much more honeys can have been studied. The results show that our values are within those already reported, with the exception of superoxide anion free radical which our results are much better than those reported by Aazza et al. [12]. The enzyme inhibitory activities were for the first time reported by this team [13] for one sample of *E. officinarum* (M3) and one sample of *E. resinifera* (M2) which are completed in the present work with other *Euphorbia* honey samples. In all cases, the phenolic fraction of all honeys seem to have a role in all activities found, since in all cases the activities were better than the entire honeys, although in some cases, a positive correlation between the total phenol of the honey extract and the activities were found, meaning these results that beyond the total phenolic content, the type of compound is also important.

## 5. Conclusions

All honey samples were found to be within the acceptable limit of the international standards. *E. resinifera* honeys presented lower moisture, HMF but higher reducing sugar percentage than *E. officinarum* honeys. The mineral analyses showed that the main dominant mineral element in the two *Euphorbia* honey types is potassium (K). Phenolic compounds gallic acid, 4-hydroxybenzoic acid and *p*-coumaric acid were detected in all extracts, although in different ratios, whereas kampferol was only present in two samples. The methanolic extracts of the honey recorded the highest values in all the biological activities, in comparison with the corresponding entire honey. Moreover, *E. resinifera* honeys had better anti-inflammatory activities than the *E. officinarum* honeys, while in the remaining biological activities it was not possible to observe differences between the two monofloral honey types. To conclude, this study revealed that *Euphorbia* honey may be a source of antioxidant molecules. In addition, it may provide new compounds useful for the symptom treatment of various diseases (e.g., gout, hyperpigmentation or Alzheimer) or delay its progression.

## Figures and Tables

**Figure 1 foods-10-01909-f001:**
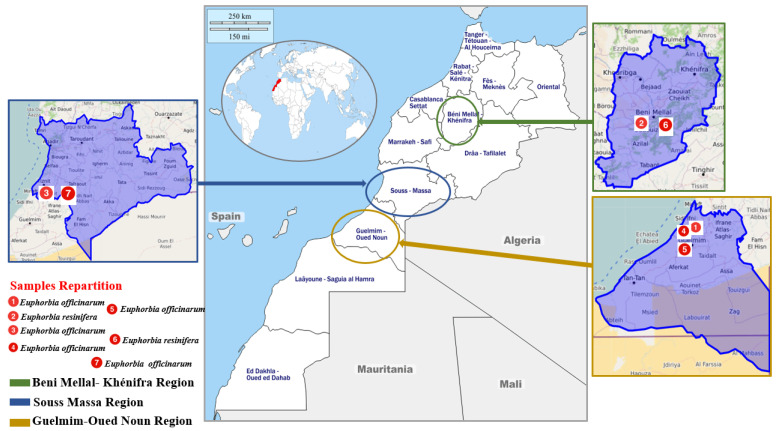
Map showing the location of the apiaries in which samples were collected.

**Figure 2 foods-10-01909-f002:**
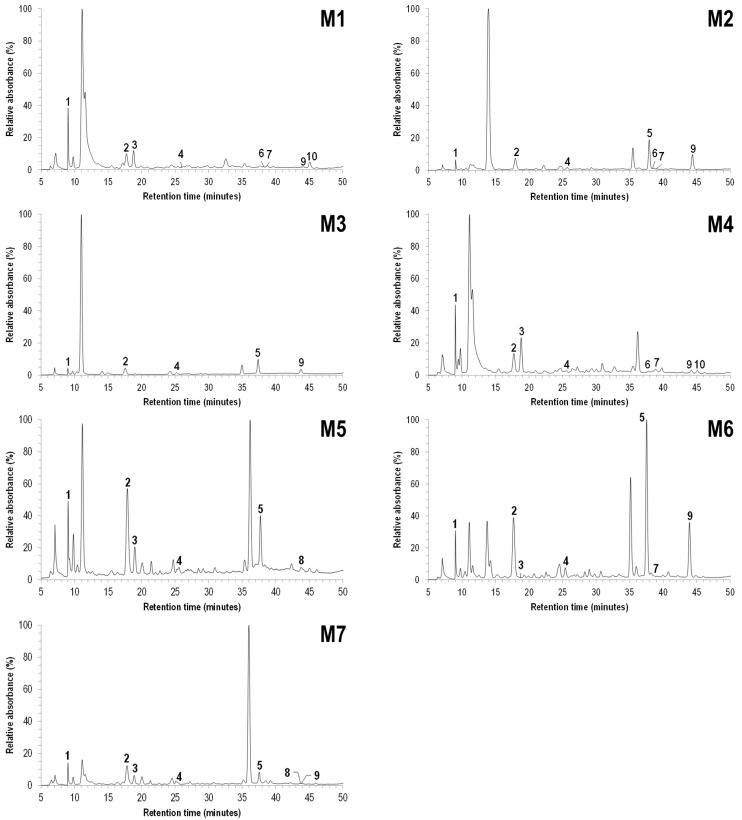
LC/DAD chromatographic profiles of the studied honey extracts, recorded at 270 nm. Peak identification in Table 6.

**Table 1 foods-10-01909-t001:** Honey samples, place, and the most predominant pollen of seven *Euphorbia* honey samples from Morocco. The results of M2 and M3 samples were previously published [13]. Reproduced with permission from Oumaima Boutoub et al., Antioxidant activity and enzyme inhibitory potential of Euphorbia resinifera and E. officinarum honeys from Morocco and plant aqueous extracts; published by Springer Nature, 2020.

Honey/Region of Production/Coordinates	Pollen Species (%)
**M1/** Guelmim-Oued Noun/28°27′ N, 10°07′ W	*E. officinarum* 55.67 ± 1.78 *Quercus rotindifolia* 10.31 ± 0.35 *Genista hirsuta* 6.88 ± 079 *Thimus lotocephalus* 4.95 ± 0.2 *Cistus albidus* 4.08 ± 0.47 *Elix aquifolium* 2.97 ± 0.10 *Malus sylvestris* 2.9 ± 0.15 *Malus domestica* 2.83 ± 0.75 *Eucaliptus cinereae* 2.32 ± 0.34 *Cistus crepis* 2.1 ± 0.35 *Campanula pimulifolia* 1.64 ± 0.11 *Lavandula viridis* 1.46 ± 0.13
**M2/** Beni Mellal-Khénifra/32°22′06″ N, 6°22′09″ W	*E. resinifera* 48.7 ± 1.1 *Caesalpinia pulcherrima* 21.8 ± 1.3 *Malvus domestica* 10.2± 0.4 *Cistus crepis* 7.9 ± 0.9 *Populus nigra* 4.0 ± 0.3 *Genista hirsuta* 2.9 ± 0.2 *Populus alba* 1.9 ± 0.2 *Elix aquifolium* 2.6 ± 0.3
**M3/** Souss-Massa/30°04′48″ N, 8°28′48″ W	*E. officinarum* 52.1 ± 1.6 *Caesalpinia pulcherrima* 11.8± 0.7 *Arbutus unedo* 6.1 ± 1.2 *Populus alba* 5.8 ± 0.8 *Pinus pinaster* 5.0 ± 0.2 *Eucalyptus globulus* 3.3 ± 0.6 *Malvus domestica* 3.0 ± 0.2 *Thymus lotocephalus* 2.4 ± 0.3 *Quercus suber* 2.0 ± 0.1 *Eucalyptus cinereae* 1.9 ± 0.2 *Populus nigra* 1.8 ± 0.3 *Caesalpinia spinosa* 1.7 ± 0.1 *Cistus albidus* 1.7 ± 0.2 *Trifolium arvense* 1.5 ± 0.3
**M4/** Guelmim-Oued Noun/28°27′ N, 10°07′ W	*E. officinarum* 51.78 ± 2.11 *Caesalpinia spinosa* 14.95 ± 1.84 *Arbustus unedo* 9.93 ± 0.61 *Cistus crepis* 9.89 ± 0.57 *Cistus populis* 5.62 ± 0.61 *Eucaliptus globulus* 5.01 ± 0.45 *Populus alva* 2.84 ± 0.39
**M5/** Guelmim-Oued Noun/28°27′ N, 10°07′ W	*E. officinarum* 51.51 ± 2.53 *Quercus suber* 17.74 ± 2.01 *Quercus rotindifolia* 10.34 ± 0.11 *Caesalpinia spinosa* 6.91 ± 0.35 *Olea europaceae* 4.81 ± 0.27 *Trifolium arvenses* 4.32 ± 0.35 *Populus alva* 2.88 ± 0.21 *Malus domestica* 1.51 ± 0.06
**M6/** Beni Mellal-Khénifra/32°22′06″ N, 6°22′09″ W	*E. resinifera* 45.58 ± 1.98 *Genista hirsuta* 12.83 ± 0.38 *Salvia officinalis* 6.06 ± 0.72 *Euphorbia officinarum* 5.93 ± 0.15 *Cistus populis* 5.35 ± 0.68 *Malus domestica* 5.11 ± 0.33 *Cistus crepis* 3.22 ± 0.25 *Cistus albidus* 4.22 ± 0.33 *Quercus rotindifolia* 4.17 ± 0.39 *Campanula pimulifolia* 2.96 ± 0.07 *Populus alva* 2.40 ± 0.17 *Malus sylvestris* 2.17 ± 0.39
**M7/** Souss-Massa/30°04′48″ N, 8°28′48″ W	*E. officinarum* 44.22 ± 4.57 *Pinus pinaster* 14.16 ± 1.82 *Caesalpinia pulcherrima* 6.33 ± 0.35 *Malus domestica* 6.22 ± 0.35 *Quercus suber* 4.99 ± 0.24 *Eucaliptus globulus* 4.73 ± 0.73 *Caesalpinia spinosa* 4.35 ± 0.9 *Cistus crepis* 3.62 ± 0.16 *Artenisa vulgaris* 3.15 ± 0.72 *Salvia officinalis* 2.5 ± 0.15 *Asparagus albus* 2.05 ± 0.27 *Populus nigra* 1.90 ± 0.12 *Lavandula viridis* 1.81 ± 0.24

**Table 2 foods-10-01909-t002:** Physicochemical parameters of Moroccan Euphorbia honeys. The results of M2 and M3 samples were already reported [13]. Reproduced with permission from Oumaima Boutoub et al., Antioxidant activity and enzyme inhibitory potential of Euphorbia resinifera and *E. officinarum* honeys from Morocco and plant aqueous extracts; published by Springer Nature, 2020.

	pH	Moisture (%)	Diastase (Shade units/g)	Proline (mg/kg)	Conductivity (µS/cm)	Ash (%)	HMF (mg/kg)	Colour (mm)	Free Acidity (meq/kg)	Lactonic Acidity (meq/kg)	Total Acidity (meq/kg)	Reducing Sugar (%)
**M1**	3.88 ± 0.09 ^bc^	19.63 ± 0.00 ^b^	11.93 ± 1.23 ^e^	584.96 ± 42.05 ^d^	553.67 ± 0.57 ^a^	0.17 ± 0.00 ^a^	51.74 ± 0.01 ^b^	136.84 ± 0.48 ^f^ **Dark Amber**	15.68 ± 1.75 ^b^	7.84 ± 2.92 ^c^	23.52 ± 2.88 ^bc^	61.67 ± 0.00 ^c^
**M2**	4.08 ± 0.04 ^a^	18.62 ± 0.00 ^e^	37.36 ± 1.46 ^d^	953.94 ± 36.50 ^bc^	379.33 ± 0.57 ^f^	0.14 ± 0.00 ^c^	2.29 ± 0.00 ^d^	407.59 ± 0.84 ^b^ **Dark Amber**	10.08 ± 1.83 ^d^	7.68 ± 1.25 ^c^	17.76 ± 1.44 ^d^	66.67 ± 0.00 ^b^
**M3**	4.06 ± 0.02 ^a^	19.00 ± 0.00 ^cd^	13.19 ± 1.33 ^e^	729.56 ± 43.27 ^cd^	342.33 ± 1.52 ^g^	0.14 ± 0.00 ^bc^	80.42 ± 0.13 ^a^	294.68 ± 1.28 ^c^ **Dark Amber**	10.64 ± 2.04 ^cd^	10.96 ± 0.91 ^c^	21.60 ± 2.19 ^cd^	61.67 ± 0.01 ^c^
**M4**	3.99 ± 0.02 ^ab^	19.06 ± 0.00 ^c^	11.43 ± 1.20 ^e^	692.48 ± 26.69 ^d^	449.00 ± 1.00 ^d^	0.17 ± 0.00 ^ab^	41.02 ± 0.14 ^c^	191.56 ± 0.84 ^d^ **Dark Amber**	12.60 ± 0.37 ^c^	10.44 ± 1.61 ^c^	23.04 ± 0.83 ^bc^	61.67 ± 0.00 ^c^
**M5**	3.84 ± 0.01 ^cd^	20.00 ± 0.00 ^a^	49.61 ± 0.75 ^c^	1169.45 ± 18.9 ^b^	514.00 ± 1.00 ^b^	0.19 ± 0.00 ^a^	7.04 ± 0.14 ^d^	144.90 ± 0.84 ^e^ **Dark Amber**	19.04 ± 0.77 ^a^	15.60 ± 0.43 ^b^	34.08 ± 0.83 ^a^	61.00 ± 0.01 ^c^
**M6**	4.01 ± 0.01 ^a^	18.73 ± 0.00 ^de^	115.89 ± 1.77 ^a^	1485.71 ± 23.6 ^a^	455.33 ± 0.57 ^c^	0.18 ± 0.00 ^a^	2.40 ± 0.00 ^d^	510.96 ± 0.48 ^a^ **Dark Amber**	17.36 ± 0.37 ^ab^	20.56 ± 1.51 ^a^	37.92 ± 1.44 ^a^	70.67 ± 0.00 ^a^
**M7**	3.75 ± 0.01 ^d^	19.13 ± 0.00 ^c^	103.23 ± 2.42 ^b^	962.80 ± 44.43 ^bc^	403.67 ± 0.57 ^e^	0.13 ± 0.00 ^c^	5.89 ± 0.00 ^d^	103.67 ± 0.96 ^g^ **Amber**	19.04 ± 1.54 ^a^	8.32 ± 0.69 ^c^	27.36 ± 1.66 ^b^	61.00 ± 0.01 ^c^

The values in the same column followed by the same letter are not significantly different (*p* < 0.05) by Tukey’s multiple range test.

**Table 3 foods-10-01909-t003:** Sugar content (g/100 g) of *Euphorbia resinifera* and *Euphorbia officinarum* monofloral honey samples from Morocco. The results of M2 and M3 samples were previously published [13]. Reproduced with permission from Oumaima Boutoub et al., Antioxidant activity and enzyme inhibitory potential of Euphorbia resinifera and E. officinarum honeys from Morocco and plant aqueous extracts; published by Springer Nature, 2020.

	Fructose	Glucose	Sucrose	Turanose	Maltose	Trehalose
**M1**	36.40 ± 3.10 ^a^	32.31 ± 2.70 ^a^	5.37 ± 0.13 ^a^	5.40 ± 0.59 ^a^	3.21 ± 0.40 ^ab^	4.58± 0.50 ^a^
**M2**	37.02 ± 2.11 ^a^	34.17 ± 1.78 ^a^	4.09 ± 0.33 ^a^	2.11 ± 0.50 ^b^	2.26 ± 0.42 ^bc^	2.80 ± 0.52 ^bc^
**M3**	34.95 ± 1.08 ^a^	30.23 ± 1. 80 ^a^	4.27 ± 0.90 ^a^	2.83 ± 0.55 ^b^	3.79 ± 0.47 ^a^	3.98 ± 0.53 ^ab^
**M4**	33.04 ± 1.18 ^a^	29.15 ± 1.33 ^a^	3.45 ± 0.91 ^a^	1.98 ± 0.49 ^b^	1.89 ± 0.44 ^bc^	1.70 ± 0.50 ^cd^
**M5**	39.48 ± 1.11 ^a^	31.53 ± 2.30 ^a^	5.04 ± 0.91 ^a^	2.57 ± 0.41 ^b^	1.75 ± 0.48 ^c^	1.04 ± 0.51 ^d^
**M6**	31.53 ± 1.33 ^a^	27.77 ± 2.11 ^a^	4.23 ± 0.94 ^a^	1.28 ± 0.49 ^b^	1.58 ± 0.47 ^c^	0.62 ± 0.51 ^c^
**M7**	33.79 ± 3.02 ^a^	30.85 ± 2.83 ^a^	4.36 ± 0.90 ^a^	2.36 ± 0.50 ^b^	2.49 ± 0.49 ^abc^	1.63 ± 0.49 ^cd^

The values in the same column followed by the same letter are not significantly different (*p* < 0.05) by Tukey’s multiple range test.

**Table 4 foods-10-01909-t004:** Mineral content (mg/kg) in Moroccan *Euphorbia* honeys. *E. officinarum* (samples M1, M3, M4, M5, M7) and *E. resinifera* (samples M2, M6). * The results of M2 and M3 samples were previously published [13]. Reproduced with permission from Oumaima Boutoub et al., Antioxidant activity and enzyme inhibitory potential of Euphorbia resinifera and E. officinarum honeys from Morocco and plant aqueous extracts; published by Springer Nature, 2020.

	Fe	Zn	Mn	Cu	Al	Ca	K	Mg	Na
**M1**	5.79 ± 0.01	3.30 ± 0.24 ^a^	0.81 ± 0.00 ^d^	<LOD	9.35 ± 0.41	152.72 ± 2.51 ^a^	352.48 ± 5.74 ^c^	55.80 ± 0.07 ^a^	125.05 ± 1.90 ^a^
**M2**	10.27 ± 0.21 ^b^	6.29 ± 0.21 ^a^	1.08 ± 0.00 ^c^	<LOD	11.39 ± 1.34 ^c^	114.82 ± 1.55 ^b^	344.82 ± 13.25 ^c^	39.63 ± 1.02 ^b^	41.92 ± 0.10 ^c^
**M3**	332.47 ± 14.46 ^a^	1.77 ± 0.16 ^a^	1.46 ± 0.26 ^b^	109.68 ± 3.67 ^a^	64.25 ± 9.54 ^a^	70.93 ± 1.48 ^e^	410.22 ± 0.17 ^b^	32.72 ± 3.04 ^c^	35.20 ± 3.64 ^c^
**M4**	6.68 ± 0.02 ^b^	2.14 ± 0.04 ^a^	1.09 ± 0.01 ^c^	<LOD	12.45 ± 3.95 ^bc^	119.83 ± 3.42 ^b^	396.80 ± 8.13 ^b^	32.88 ± 1.23 ^c^	87.30 ± 2.11 ^b^
**M5**	6.19 ± 0.57 ^b^	6.21 ± 0.94 ^a^	0.91 ± 0.01 ^cd^	<LOD	9.23 ± 1.80 ^c^	94.00 ± 1.62 ^d^	533.15 ± 7.28 ^a^	22.60 ± 0.49 ^d^	34.96 ± 4.38 ^c^
**M6**	12.92 ± 3.56	2.5 ± 0.17 ^a^	1.71 ± 0.03 ^a^	<LOD	19.90 ± 3.39 ^b^	102.57 ± 1.34 ^cd^	502.48 ± 3.85 ^a^	31.93 ± 0.81 ^c^	44.43 ± 1.52 ^c^
**M7**	9.20 ± 0.28 ^b^	1.97 ± 0.17 ^a^	0.75 ± 0.02 ^d^	<LOD	6.32 ± 0.03 ^c^	105.75 ± 1.06 ^c^	323.95 ± 2.13 ^c^	31.20 ± 0.49 ^c^	42.25 ± 1.13 ^c^

The values in the same column followed by the same letter are not significantly different (*p* < 0.05) by Tukey’s multiple range test. LOD (limit of detection) = 0.786 mg/mL.

**Table 5 foods-10-01909-t005:** Total phenols’ content of *Euphorbia* honey samples (M1, M2, M3, M4, M5, M6, M7) and their phenolic extracts (PE1, PE2, PE3, PE4, PE5, PE6, PE7). The results of M2 and M3 samples were previously published [13] Reproduced with permission from Oumaima Boutoub et al., Antioxidant activity and enzyme inhibitory potential of Euphorbia resinifera and E. officinarum honeys from Morocco and plant aqueous extracts; published by Springer Nature, 2020.

Total Phenols’ Content of Honey Samples (mg GAE/100 g)		Total Phenols’ Content of Phenolic Extract (mg GAE/100 g)	
**M1**	64.78 ± 0.02 ^ab^	**PE1**	4.74 ± 0.00 ^d^
**M2**	54.55 ± 0.02 ^c^	**PE2**	5.93 ± 0.00 ^d^
**M3**	61.82 ± 0.03 ^b^	**PE3**	10.24 ± 0.01 ^c^
**M4**	53.38 ± 0.02 ^c^	**PE4**	13.88 ± 0.00 ^b^
**M5**	64.94 ± 0.05 ^ab^	**PE5**	30.74 ± 0.00 ^a^
**M6**	69.25 ± 0.01 ^a^	**PE6**	13.55 ± 0.01 ^b^
**M7**	46.14 ± 0.03 ^d^	**PE7**	12.52 ± 0.00 ^b^

The values in the same column followed by the same letter are not significantly different (*p* < 0.05) by Tukey’s multiple range test.

**Table 6 foods-10-01909-t006:** List of compounds identified in each honey sample, using LC/MS analysis.

Compound	R_t_ (min)	[M–H] ^-^	SRM *	M1	M2	M3	M4	M5	M6	M7
Gallic acid (1)	9.01	169	169→125	†	†	†	†	†	†	†
4-Hydroxybenzoic acid (2)	17.89	137	137→93	†	†	†	†	†	†	†
Caffeic acid (3)	18.99	179	179→135	†			†	†	†	†
*p*-Coumaric acid (4)	25.42	163	163→119	†	†	†	†	†	†	†
Abscisic acid ^†^ (5)	37.54	263	-		†	†		†	†	†
Luteolin (6)	38.12	285	285→241	†	†		†			
Quercetin (7)	38.77	301	301→179	†	†		†		†	
Apigenin (8)	43.81	269	269→149					†		†
Naringenin (9)	44.38	271	271→151	†	†	†	†		†	†
Kaempferol (10)	45.08	285	285→185	†			†			

* Selective Reaction Monitoring (SRM) at a normalized collision energy of 30%. † Abscisic acid identified using full scan mode.

**Table 7 foods-10-01909-t007:** Data presenting the different antioxidant activity (DPPH, NO and superoxyde inhibition) of *Euphorbia* honey samples (M1, M2, M3, M4, M5, M6, M7) and their own phenolic extract (PE1, PE2, PE3, PE4, PE5, PE6, PE7). The results of M2 and M3 samples were previously published [13]. Reproduced with permission from Oumaima Boutoub et al., Antioxidant activity and enzyme inhibitory potential of Euphorbia resinifera and E. officinarum honeys from Morocco and plant aqueous extracts; published by Springer Nature, 2020.

Honey Samples	Antioxidant Activity IC_50_ (mg/mL)		
	DPPH	NO	Superoxide
**M1**	52.30 ± 1.74 ^b^	85.67 ± 0.41 ^d^	3.86 ± 0.10 ^b^
**M2**	80.13 ± 1.11 ^a^	88.20 ± 0.78 ^d^	3.71 ± 0.01 ^b^
**M3**	55.48 ± 0.73 ^b^	115.58 ± 1.35 ^a^	2.82 ± 0.15 ^c^
**M4**	38.23 ± 0.31 ^c^	111.29 ± 2.91 ^b^	4.26 ± 0.09 ^a^
**M5**	16.30 ± 0.41 ^d^	76.24 ± 0.55 ^e^	2.75 ± 0.00 ^c^
**M6**	78.50 ± 2.00 ^a^	94.95 ± 1.99 ^c^	2.71 ± 0.01 ^c^
**M7**	75.83 ± 3.63 ^a^	116.48 ± 0.36 ^a^	1.95 ± 0.02 ^d^
**Phenolic Extracts**	**DPPH**	**NO**	**Superoxide**
**PE1**	17.16 ± 0.18 ^b^	20.13 ± 0.19 ^f^	1.08 ± 0.01 ^e^
**PE2**	10.24 ± 0.13 ^d^	21.16 ± 0.36 ^e^	0.95 ± 0.00 ^f^
**PE3**	17.64 ± 0.12 ^a^	3.64 ± 0.16 ^g^	4.10 ± 0.02 ^a^
**PE4**	14.77 ± 0.26 ^c^	24.66 ± 0.26 ^d^	3.39 ± 0.00 ^b^
**PE5**	2.58 ± 0.04 ^f^	34.18 ± 0.16 ^b^	1.24 ± 0.00 ^d^
**PE6**	17.54 ± 0.12 ^ab^	27.82 ± 0.20 ^c^	1.78 ± 0.05 ^c^
**PE7**	4.21 ± 0.13 ^e^	37.87 ± 0.24 ^a^	1.09 ± 0.01 ^e^

The values in the same column for each sample group followed by the same letter are not significantly different (*p* < 0.05) by Tukey’s multiple range test

**Table 8 foods-10-01909-t008:** Data presenting the different enzymatic activities (Acetylcholinesterase, lipoxygenase, tyrosinase and xanthine oxidase inhibition) of *Euphorbia* honey samples (M1, M2, M3, M4, M5, M6, M7) and their own phenolic extract (PE1, PE2, PE3, PE4, PE5, PE6, PE7). The results of M2 and M3 samples were previously published [13]. Reproduced with permission from Oumaima Boutoub et al., Antioxidant activity and enzyme inhibitory potential of Euphorbia resinifera and E. officinarum honeys from Morocco and plant aqueous extracts; published by Springer Nature, 2020.

Honeys Samples	Enzymatic Activities IC_50_ (mg/mL)			Percentage Inhibition (%)
	Acetylcholinesterase	Lipoxygenase	Tyrosinase	Xanthine Oxidase
**M1**	154.98 ± 5.75 ^a^	48.85 ± 0.63 ^b^	82.61 ± 6.35 ^b^	85.27 ± 0.31 ^bc^
**M2**	44.65 ± 8.33 ^b^	32.72 ± 0.35 ^e^	11.46 ± 1.80 ^e^	71.74 ± 1.89 ^d^
**M3**	164.49 ± 8.50 ^a^	46.77 ± 0.39 ^c^	54.90 ± 3.23 ^c^	94.92 ± 0.39 ^a^
**M4**	16.77 ± 0.34 ^c^	43.01 ± 0.06 ^d^	97.06 ± 5.89 ^a^	84.75 ± 0.27 ^c^
**M5**	4.52 ± 0.88 ^c^	43.43 ± 1.22 ^d^	39.49 ± 1.32 ^d^	66.34 ± 1.03 ^e^
**M6**	16.30 ± 1.56^c^	29.04 ± 0.33 ^f^	67.27 ± 3.16 ^c^	87.66 ± 0.66 ^b^
**M7**	3.90 ± 0.60 ^c^	51.54 ± 0.20 ^a^	42.21 ± 6.25 ^d^	48.78 ± 0.73 ^f^
**Phenolic** **Extracts**	**Acetylcholinesterase**	**Lipoxygenase**	**Tyrosinase**	**Xanthine Oxidase**
**PE1**	1.35 ± 0.01 ^d^	0.43 ± 0.00 ^c^	16.27 ± 0.03 ^d^	97.69 ± 0.38 ^b^
**PE2**	7.76 ± 0.51 ^bc^	0.32 ± 0.00 ^f^	15.28 ± 0.17 ^d^	96.54 ± 0.38 ^c^
**PE3**	8.09 ± 0.04 ^b^	0.50 ± 0.00 ^b^	40.15 ± 0.60 ^b^	99.66 ± 0.08 ^a^
**PE4**	13.62 ± 0.22 ^a^	0.51 ± 0.00 ^a^	44.71 ± 0.00 ^a^	96.15 ± 0.08 ^c^
**PE5**	1.13 ± 0.06 ^d^	0.51 ± 0.00 ^a^	37.89 ± 1.21 ^bc^	90.91 ± 0.63 ^e^
**PE6**	13.32 ± 0.32 ^a^	0.38 ± 0.00 ^d^	41.13 ± 2.86 ^b^	94.76 ± 0.08 ^d^
**PE7**	7.10 ± 0.32^c^	0.36 ± 0.00 ^e^	35.76 ± 0.29 ^c^	48.56 ± 0.22 ^f^

The values in the same column for each sample group followed by the same letter are not significantly different (*p* < 0.05) by Tukey’s multiple range test.

**Table 9 foods-10-01909-t009:** The correlations between the results of TPC, antioxidant activities (DPPH, NO, superoxyde scavenging activities) and inhibition of acetylcholinesterase, lipoxygenase and tyrosinase activities of *Euphorbia* honey samples and *Euphorbia* honey phenolic extract.

	*Euphorbia* Honeys Samples						
Correlation	TPC	DPPH	NO	Superoxide	ACTE	Lipoxygenase	Tyrosinase
**TPC**	1	−0.23	−0.56 **	0.07	0.30	−0.37	0.22
**DPPH**	−0.23	1	0.31	−0.23	0.01	−0.36	−0.30
**NO**	−0.56 **	0.31	1	−0.22	0,08	0.35	0.24
**Superoxide**	0.06	−0.23	−0.22	1	0.25	−0.17	0.41
**ACTE**	0.30	0.01	0.08	0.25	1	0.34	0.20
**Lipoxygenase**	−0.37	−0.36	0.35	−0.17	0.34	1	0.22
**Tirosyanse**	0.22	−0.30	0.24	0.41	0.20	0.22	1
	***Euphorbia* Honey Phenolic Extract**						
**TPC**	1	−0.61 **	−0.50 *	−0.04	−0.22	0.50 *	0.56 **
**DPPH**	−0.61 **	1	−0.72 **	−0.53 *	0.43	0.13	−0.13
**NO**	−0.49 *	−0.719 **	1	−0.66 **	−0,01	−0.27	0.14
**Superoxide**	−0.04	0.53 *	−0.66**	1	0.5 *	0.66 **	0.57 **
**ACTE**	−0.22	0.43	−0.01	0.5 *	1	−0.09	0.57 **
**Lipoxygenase**	0.50 *	0.13	−0.28	0.65 **	−0.09	1	0.42
**Tyrosinase**	0.56 **	−0.13	0.14	0.57 **	0.58 **	0.42	1

** Correlation is significant at the *p* < 0.01 level. * Correlation is significant at the *p* < 0.05 level.

**Table 10 foods-10-01909-t010:** IC_50_ values (mg/mL) found for the assays DPPH, NO and superoxide found in other works on Moroccan honeys and those found in the present work.

Honey Type	DPPH	NO	Superoxide	Reference
*Thymus* spp	ND	21.47 95.14	ND	[12]
*Euphorbia resinifera*				
*Bupleurum spinosum*	15.34	125.89	50.91	[35]
*Ceratonia siliqua*	12.54–23.52	ND	ND	[44]
*Thymus vulgaris*	5.57	ND	ND	[45]
*Peganum harmala*	48.67			
*E. resinifera*	15.34			
*Bupleurum spinosum*	13.57–45.34	ND	ND	[46]
*E. officinarum*	16.30–75.83	76.24–116.48	1.95–4.26	Present work
*Euphorbia resinifera*	78.50–80.13	88.20–94.95	2.71–3.71	Present work

## Data Availability

Not applicable.

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
