# Peer review of "Comparative Study of the Antioxidant and Enzyme Inhibitory Activities of Two Types of Moroccan Euphorbia Entire Honey and Their Phenolic Extracts"

_foods, 2021, doi:10.3390/foods10081909_

Round 1

Reviewer 1 Report

This Manuscript entitled " Comparative study of the antioxidant and enzyme inhibitory activities of two types of Moroccan Euphorbia entire honey and their phenolic extracts “ for Foods Journal aims to evaluate the most important biological activities, such as antioxidant, anti-inflammatory, anti-acetylcholinesterase, anti-tyrosinase and anti-xanthine oxidase of seven entire Euphorbia unifloral honeys of E. resinifera and E. of-ficinarum origins as well as their phenolic extracts and tentatively identify, through liquid chromatography coupled to diode array detection and mass spectrometry (LC/DAD/MS) the main phenolic compounds.

Regarding this manuscript it is well written paper but is more likely to professional paper than scientific paper. So please be so kind and in re-writing the manuscript please stress the importance of the work, how it advances current knowledge and what new science it contributes to the field of study.  

The work described should be innovative either in the approach or in the methods used. Many studies about the analysis of honey are published in the world. And according to that improvement of introduction part is needed.

Author Response

Dear Editor,

We thank you for your e-mail with the comments on our manuscript Ref.:  Manuscript ID: foods-1307804 " Comparative study of the antioxidant and enzyme inhibitory activities of two types of Moroccan Euphorbia entire honey and their phenolic extracts". We have read it carefully and we do understand and acknowledge your comments. All corrections in the manuscript text was done by using the track changes tool of the Microsoft Word.

REVIEWER 1

Comments and Suggestions for Authors

This Manuscript entitled " Comparative study of the antioxidant and enzyme inhibitory activities of two types of Moroccan Euphorbia entire honey and their phenolic extracts “ for Foods Journal aims to evaluate the most important biological activities, such as antioxidant, anti-inflammatory, anti-acetylcholinesterase, anti-tyrosinase and anti-xanthine oxidase of seven entire Euphorbia unifloral honeys of E. resinifera and E. of-ficinarum origins as well as their phenolic extracts and tentatively identify, through liquid chromatography coupled to diode array detection and mass spectrometry (LC/DAD/MS) the main phenolic compounds.

Regarding this manuscript it is well written paper but is more likely to professional paper than scientific paper. So please be so kind and in re-writing the manuscript please stress the importance of the work, how it advances current knowledge and what new science it contributes to the field of study.  

The work described should be innovative either in the approach or in the methods used. Many studies about the analysis of honey are published in the world. And according to that improvement of introduction part is needed.

Response: We acknowledge the suggestion and we have re-written the Introduction, focusing on the importance of this work developed by this team.

We hope that we have addressed correctly the remarks.

Hoping that everything is in the correct form and looking forward to hearing from you.

Yours sincerely,

Graça Miguel

Reviewer 2 Report

The aim of this research article was to evaluate the antioxidant and enzyme inhibitory activities of Euphorbia honey obtained from Morocco. Though the research is very methodical and a comprehensive analysis has been performed on the honey samples, I feel that there is something still lacking from the manuscript. At the moment, the manuscript just details comprehensive analysis of the honey samples. To conclude the work, instead of summarising all results obtained from the study, suggestions detailing the application of these honey samples across the pharmaceutical/cosmetic industry is needed (or perhaps a separate section prior to the conclusion). I cannot fault the experimental analysis, but I do feel the results need to be discussed in the wider context of their potential applications. Above all, the paper is well written and planned, I would just encourage the authors to go over the manuscript and check the English as there are very few grammatical errors that need addressing.

Points to address:

  1. In the introduction section, perhaps a better and wider explanation of the significance of this study is needed. As currently written, the rationale behind the study is clear, but there needs to be greater emphasis on the actual significance of the study i.e what is the current use of synthetic agents/precursors across industry, why are these bad, how will identifying natural agents be of greater benefit, how much could the contribute to the economy, are there any government policies or incentives that are promoting this change etc?
  2. There is too much methodological detail in the introduction – stating that phenolic compounds were identified ‘through liquid chromatography coupled to diode array detection and mass spectrometry (LC/DAD/MS)’ is not really needed.
  3. To make the introduction stronger, perhaps conclude by stating the importance of this study and how the findings from this research will contribute towards the research field.
  4. Section 2.3 – discuss the sample preparation for the physico-chemical parameters first, followed by extract preparation for phenolics. This is because in section 2.3, it has been written that physico-chemical preparation would be conducted first, then phenolic compound preparation. This would make the section flow better.
  5. Section 2.4 – please provide more detail to these methods listed that were followed.
  6. Section 2.5 – please put the equation used on a separate line.
  7. The results and discussions section is extremely lengthy. Consider making these sections more concise – not all results need to be discussed in so much detail without explaining their wider application.
  8. Please include a section prior to the conclusion section where the findings can be compared to published findings in the literature, and discuss the significance of the results obtained.

Author Response

Dear Editor,

We thank you for your e-mail with the comments on our manuscript Ref.:  Manuscript ID: foods-1307804 " Comparative study of the antioxidant and enzyme inhibitory activities of two types of Moroccan Euphorbia entire honey and their phenolic extracts". We have read it carefully and we do understand and acknowledge your comments. All corrections in the manuscript text was done by using the track changes tool of the Microsoft Word.

REVIEWER 2

Comments and Suggestions for Authors

The aim of this research article was to evaluate the antioxidant and enzyme inhibitory activities of Euphorbia honey obtained from Morocco. Though the research is very methodical and a comprehensive analysis has been performed on the honey samples, I feel that there is something still lacking from the manuscript. At the moment, the manuscript just details comprehensive analysis of the honey samples. To conclude the work, instead of summarising all results obtained from the study, suggestions detailing the application of these honey samples across the pharmaceutical/cosmetic industry is needed (or perhaps a separate section prior to the conclusion). I cannot fault the experimental analysis, but I do feel the results need to be discussed in the wider context of their potential applications. Above all, the paper is well written and planned, I would just encourage the authors to go over the manuscript and check the English as there are very few grammatical errors that need addressing.

Points to address:

  1. In the introduction section, perhaps a better and wider explanation of the significance of this study is needed. As currently written, the rationale behind the study is clear, but there needs to be greater emphasis on the actual significance of the study i.e what is the current use of synthetic agents/precursors across industry, why are these bad, how will identifying natural agents be of greater benefit, how much could the contribute to the economy, are there any government policies or incentives that are promoting this change etc?

Response: The Introduction was re-written focusing on the economic importance of this product in Morocco, reviewing what has been investigated in biological attributes terms of Euphorbia honey and according to this current knowledge how our research can contribute to unravel other possible biological features and which honey parts such properties can be attributed.

There is too much methodological detail in the introduction – stating that phenolic compounds were identified ‘through liquid chromatography coupled to diode array detection and mass spectrometry (LC/DAD/MS)’ is not really needed.

Response: This was eliminated

  1. To make the introduction stronger, perhaps conclude by stating the importance of this study and how the findings from this research will contribute towards the research field.

Response: The Introduction was re-written, we hope that this new Introduction could illustrate much better the goal of our research.

  1. Section 2.3 – discuss the sample preparation for the physico-chemical parameters first, followed by extract preparation for phenolics. This is because in section 2.3, it has been written that physico-chemical preparation would be conducted first, then phenolic compound preparation. This would make the section flow better.

Response: Thank you for this suggestion, in fact this sequence is much better and it was done.

  1. Section 2.4 – please provide more detail to these methods listed that were followed.

Response: As requested a more detailed methodology was introduced.

  1. Section 2.5 – please put the equation used on a separate line.

Response: As requested the equation is on a separate line.

  1. The results and discussions section is extremely lengthy. Consider making these sections more concise – not all results need to be discussed in so much detail without explaining their wider application.

Response: The results and discussion was shorten as suggested.

  1. Please include a section prior to the conclusion section where the findings can be compared to published findings in the literature, and discuss the significance of the results obtained.

Response: This section was introduced but only considering Moroccan honeys.

Concerning the repetition of our manuscript is 31%, after iThenticate report, some of them are impossible to change since they are not possible to change:

59 – Author’s institutional address: not possible to change, it’s not a repetition

31 – Author’s institutional web domain and author’s name initials: not possible to change, it’s not a repetition

42 – Author’s institutional address: not possible to change, it’s not a repetition

6 – Technique’s name and abbreviation: not possible to change, it’s not a repetition

14 – Samples’ names: not possible to change, it’s not a repetition

40 – List of identified compounds and their ID number in the chromatogram: not possible to change, it’s not a repetition.

There are other parts of the manuscript that were modified, but there are other ones that cannot be modified since there are results already published in the reference 13 (Table of pollen analysis), now this indication is provided. They were introduced because the phenolic extracts as well as the chemical composition is now given in this manuscript.

We hope that we have addressed correctly the remarks.

Hoping that everything is in the correct form and looking forward to hearing from you.

Yours sincerely,

Graça Miguel